# Leptin-Mediated Induction of IL-6 Expression in Hofbauer Cells Contributes to Preeclampsia Pathogenesis

**DOI:** 10.3390/ijms25010135

**Published:** 2023-12-21

**Authors:** Asli Ozmen, Chinedu Nwabuobi, Zhonghua Tang, Xiaofang Guo, Kellie Larsen, Seth Guller, Jacqueline Blas, Monica Moore, Umit A. Kayisli, Charles J. Lockwood, Ozlem Guzeloglu-Kayisli

**Affiliations:** 1Department of Obstetrics & Gynecology, Morsani College of Medicine, University of South Florida, Tampa, FL 33612, USA; asliozmen@usf.edu (A.O.); chinedu.nwabuobi@orlandohealth.com (C.N.); xiaofang@usf.edu (X.G.); klarsen@usf.edu (K.L.); blas228@usf.edu (J.B.); monicamoore@usf.edu (M.M.); uakayisli@usf.edu (U.A.K.); cjlockwood@usf.edu (C.J.L.); 2Department of Obstetrics, Gynecology and Reproductive Sciences, Yale University School of Medicine, New Haven, CT 06520, USA; zhonghua.tang@yale.edu (Z.T.); seth.guller@yale.edu (S.G.)

**Keywords:** preeclampsia, placenta, Hofbauer cells, leptin, IL-6, ERK1/2, NF-κB, JAK/STAT

## Abstract

Leptin plays a crucial role in regulating energy homoeostasis, neuroendocrine function, metabolism, and immune and inflammatory responses. The adipose tissue is a main source of leptin, but during pregnancy, leptin is also secreted primarily by the placenta. Circulating leptin levels peak during the second trimester of human pregnancy and fall after labor. Several studies indicated a strong association between elevated placental leptin levels and preeclampsia (PE) pathogenesis and elevated serum interleukin-6 (IL-6) levels in PE patients. Therefore, we hypothesized that a local increase in placental leptin production induces IL-6 production in Hofbauer cells (HBCs) to contribute to PE-associated inflammation. We first investigated HBCs-specific IL-6 and leptin receptor (LEPR) expression and compared their immunoreactivity in PE vs. gestational age-matched control placentas. Subsequently, we examined the in vitro regulation of IL-6 as well as the phosphorylation levels of intracellular signaling proteins STAT3, STAT5, NF-κB, and ERK1/2 by increasing recombinant human leptin concentrations (10 to 1000 ng/mL) in primary cultured HBCs. Lastly, HBC cultures were incubated with leptin ± specific inhibitors of STAT3 or STAT5, or p65 NF-κB or ERK1/2 MAPK signaling cascades to determine relevant cascade(s) involved in leptin-mediated IL-6 regulation. Immunohistochemistry revealed ~three- and ~five-fold increases in IL-6 and LEPR expression, respectively, in HBCs from PE placentas. In vitro analysis indicated that leptin treatment in HBCs stimulate IL-6 in a concentration-dependent manner both at the transcriptional and secretory levels (*p* < 0.05). Moreover, leptin-treated HBC cultures displayed significantly increased phosphorylation levels of STAT5, p65 NF-κB, and ERK1/2 MAPK and pre-incubation of HBCs with a specific ERK1/2 MAPK inhibitor blocked leptin-induced IL-6 expression. Our in situ results show that HBCs contribute to the pathogenesis of PE by elevating IL-6 expression, and in vitro results indicate that induction of IL-6 expression in HBCs is primarily leptin-mediated. While HBCs display an anti-inflammatory phenotype in normal placentas, elevated levels of leptin may transform HBCs into a pro-inflammatory phenotype by activating ERK1/2 MAPK to augment IL-6 expression.

## 1. Introduction

Leptin is predominantly secreted from adipose tissue and the placenta. However, its expression has also been reported in other tissues including hypothalamus, skeletal muscle, endometrium, umbilical cord, and amniotic cells [1]. Leptin binds to leptin receptors (LEPRs), which are members of the class I cytokine receptor family, and is primarily expressed by the hypothalamus, thereby regulating metabolism and appetite by inhibiting food intake, lowering body weight, and increasing metabolic rate [2]. The binding of leptin to LEPRs activates several signaling pathways, including Janus kinases (JAKs) and signal transducers and activators of transcription (STATs) [3], Nuclear factor kappa B (NF-κB), Phosphatidylinositol 3′-kinase (PI3K)-Akt (PI3K-Akt), and Mitogen-activated protein kinase (MAPK), etc. [4,5,6,7]. Thus, leptin is involved in diverse physiological processes including insulin sensitivity, angiogenesis, reproductive function, placental and fetal development, as well as immunity and inflammatory responses [1,8,9,10,11]. The placenta is the second largest leptin-producing tissue and starts leptin production as early as seven weeks of gestation [12,13]. Maternal serum leptin levels are elevated in pregnant women, with low levels seen in the first trimester, a two- to three-fold increase around 28–32 weeks of gestation, and a rapid decrease back to pre-gestational levels following delivery [2,14,15]. In addition to serving as a source of leptin [16], the human placenta is also a target of leptin actions given its high LEPRs levels [17,18,19]. The presence of both leptin and LEPRs in the placenta suggests that leptin regulates placental functions in both autocrine and paracrine manners in physiologic and pathological conditions [20].

Preeclampsia (PE) is associated with increased serum and placental leptin levels [21,22,23,24]. PE is a hypertensive disease of pregnancy, affecting approximately 6 to 8% of pregnancies in the U.S., and a leading cause of maternal and fetal morbidity and mortality [25]. As a multisystem disorder, clinical signs of PE appear during the second half of human pregnancy [26]. It is characterized by impaired extravillous trophoblasts (EVT) invasion of the decidua, leading to incomplete transformation of the spiral arteries [25,27,28]. The resulting reduced utero-placental blood flow produces placental ischemia/hypoxia [29], causing placental dysfunction, and increases placental expression of anti-angiogenic factors, proinflammatory cytokines [30,31,32,33,34], and leptin [35], contributing to the pathogenesis of PE.

Leptin increases the secretion of pro-inflammatory cytokines, in addition to its central metabolic functions. Recent studies demonstrated leptin induces secretion of interleukin (IL)-1β, IL-6, and tumor necrosis factor α (TNFα) from human term trophoblast cells [36] and human placental explants [5]. However, no data exist regarding leptin effects on Hofbauer cells (HBCs), which are resident fetal macrophages of the chorionic villi of the human placenta [37]. In normal pregnancy, HBCs exhibit mainly an anti-inflammatory macrophage (M2) phenotype, which limits inflammation and promotes angiogenesis, tissue repair, and homeostasis [38,39,40]. However, under inflammatory conditions, the M2 phenotype of HBCs switches towards the inflammatory (M1) phenotype and it produces numerous proinflammatory cytokines, e.g., IL-6, IL-1β, and TNFα [41,42]. However, the actual role of HBCs in PE is still unclear. We hypothesized that increased placental leptin expression induces IL-6 production by HBCs to contribute to PE-associated inflammation. Thus, we first investigated the expression of LEPRs and IL-6 in HBCs by immunohistochemistry in PE vs. gestational age (GA)-matched control placentas. Then, we investigated the in vitro regulation of IL-6 expression by leptin at the transcriptional, translational, and secretory levels in primary cultures of HBCs isolated from human term placenta samples.

## 2. Results

### 2.1. Elevated IL-6 and LEPR Immunoreactivity in Hofbauer Cells of PE Placentas

To investigate whether increased leptin levels in PE placentas [35,43,44] contribute to PE-associated inflammation by inducing IL-6 levels in HBCs, we initially evaluated placental IL-6 and LEPR expression in HBCs by immunohistochemistry. In the placental villi, IL-6 and LEPR immunoreactivity were observed in several cell types including syncytiotrophoblasts and cytotrophoblasts, as well as villous endothelial cells and HBCs (Figure 1). As this study focused on HBCs, immunostaining of CD68 as a monocyte/macrophage marker (Figure 1) was performed to confirm HBC identity in serial sections of placental specimens immunostained with either IL-6 or LEPR antibodies and confirmed the expression of both IL-6 and LEPR in CD68+ HBCs (Figure 1).

Later, we compared the HBCs-specific immunostaining intensity of IL-6 and LEPR in PE vs. GA-matched control placental specimens. Stronger IL-6 and LEPR immunoreactivity was observed in HBCs from PE placentas compared to GA-matched control specimens (Figure 2). No immunoreactivity was observed in negative slides. HSCORE analysis confirmed significantly increased IL-6 (Mean ± SEM; 83.0 ± 20.9 vs. 25.0 ± 4.5, *p* < 0.05) and LEPR (63.5 ± 18.6 vs. 10.9 ± 1.0, *p* < 0.05) levels in HBCs in PE vs. control placental specimens (Figure 2).

### 2.2. In Vitro Regulation of IL-6 Levels by Leptin Treatment in HBCs

To determine whether elevated leptin levels contribute to increased IL-6 expression in HBCs in PE placental specimens, we investigated *IL-6* mRNA and protein levels in primary HBC cultures treated with either vehicle (control) or human recombinant (r)-leptin (10, 100, 1000 ng/mL). Analysis of qPCR results revealed that r-leptin significantly enhanced *IL-6* mRNA levels by ~6-fold and 500-fold in cultured HBCs treated with 100 and 1000 ng/mL r-leptin (Mean ± SEM 6.6 ± 2.1 and 560.9 ± 239.6, respectively) vs. control (1.00 ± 0.001; *p* < 0.05), with no effect seen at 10 ng/mL r-leptin (Figure 3A). Secreted levels of IL-6 protein were analyzed in culture media supernatants obtained from HBCs after 24 h r-leptin treatment. ELISA analysis revealed that compared to the control treatment (0.11 ± 0.06 pg/mL), HBCs displayed increased IL-6 secretion in response to r-leptin treatment at 100 ng/mL (1.47 ± 0.8 pg/mL, *p* < 0.05) and 1000 ng/mL (60.34 ± 15.7 pg/mL, *p* < 0.05) concentrations, but not at 10 ng/mL (Figure 3B). We also analyzed intracellular IL-6 protein levels in HBCs cell lysate by immunoblotting. In contrast to increased IL-6 secretion by r-leptin, a concentration-dependent decrease in intracellular IL-6 protein levels was detected at HBC lysates exposed to 100 and 1000 ng/mL r-leptin (Figure 3C), suggesting that the mobilization of IL-6 for secretion in HBCs follows leptin exposure. In contrast, intracellular IL-6 levels were similar in the control and 10 ng/mL r-leptin-treated HBCs.

To eliminate potential residual endotoxin contamination of the r-leptin (generated in Escherichia coli, per 1000 ng r-leptin contains less than 1 ng endotoxin, detected according to data sheet of the manufacturer) on IL-6 expression, HBCs were pretreated with a lipopolysaccharide (LPS) antagonist for 1 h prior to 1000 ng/mL r-leptin or 1 ng/mL LPS [45]. qPCR results revealed that the LPS antagonist inhibited LPS-induced IL-6 expression, but not leptin-induced IL-6 expression (Appendix A), confirming the leptin-mediated-specific action on IL-6 expression in HBCs.

### 2.3. Leptin Activates STATs, NF-κB, and ERK1/2 Signaling Pathways in HBCs

Previous studies showed that leptin binding to LEPRs activates several intracellular signaling pathways, i.e., JAK/STAT, NF-κB, and ERK1/2 MAPK in a variety of cell types [46]. Therefore, to identify relevant pathways responsible for leptin-induced IL-6 expression, HBCs were treated with either control or 10, 100, 1000 ng/mL r-leptin and then subjected to immunoblotting to evaluate both the total (t-) and activated (phosphorylated; ph-) forms of STAT3, STAT5, p65 NF-κB, and ERK1/2 MAPK signaling molecules. Immunoblotting results revealed that: (1) compared to the control, only 1000 ng/mL leptin treatment significantly increased the phosphorylation of STAT5 (*p* = 0.01, Figure 4A), whereas increased STAT3 phosphorylation did not attain significance (Figure 4B); and (2) the phosphorylation of both p65 NF-κB (*p* < 0.05, Figure 4C) and ERK1/2 (*p* < 0.05, Figure 4D) was significantly increased in both 100 and 1000 ng/mL r-leptin-treated HBCs, whereas total forms of the relevant molecules were not altered by any leptin concentration.

To further identify if a cause-and-effect relationship exists between leptin regulation and the above-mentioned signaling pathways, HBCs were preincubated with specific inhibitors of these potential intracellular signaling pathways and then treated with either control or 100 or 250 or 1000 ng/mL r-leptin. During this step, we excluded 10 ng/mL r-leptin concentration and included 250 ng/mL r-leptin to the experimental setup to provide a more accurate representation of serum leptin levels reported in blood samples of women with PE [47]. Analysis of qPCR results revealed that both the STAT3 and STAT5 inhibitors significantly reduced 1000 ng/mL r-leptin-induced *IL-6* mRNA levels by 33.2% and 34.9%, respectively, in HBC culture (*p* < 0.05; Figure 5A,B). NF-κB inhibitors significantly reduced 250 ng/mL r-leptin-induced *IL-6* mRNA levels by 51.2% (*p* < 0.001, Figure 5C). However, the ERK1/2 inhibitor displayed the strongest inhibitory effect in both 250 and 1000 ng/mL r-leptin-treated HBC cultures by a ~70% reduction of *IL-6* mRNA levels (*p* < 0.001, Figure 5D).

## 3. Discussion

As a highly specialized organ, the placenta plays critical roles in facilitating nutrient, gas, and waste exchange between maternal and fetal circulations, and producing several growth factors or hormones that regulate both maternal and fetal physiology. Among these factors, placental leptin is an important cytokine that has crucial roles during pregnancy by modulating critical cellular processes, i.e., proliferation, apoptosis, and invasion in both physiologic and pathological conditions [1,18]. Increasing evidence supports the increased placental leptin production and its pathophysiological roles in PE [24,25]. It is known that PE is linked to a strong local and systemic pro-inflammatory milieu due to altered immune cell types and numbers accompanying increased proinflammatory and decreased anti-inflammatory cytokines [26]. Several studies reported that increased hypoxia, oxidative stress, and pro-inflammatory cytokines contribute to enhanced levels of leptin in the preeclamptic placenta [35,43,48,49]. HBCs refer to a heterogeneous population of fetal macrophages in the chorionic villus, and they have important functions in successful placental development [37]. Given the possible involvement of leptin in the pathophysiology of PE [24,35], we questioned how increased leptin levels alter the function of HBCs to contribute to PE-associated inflammation.

Our immunostaining results confirmed the widespread expression of LEPR in syncytiotrophoblasts, cytotrophoblasts, villous vessels, and HBCs, as previously demonstrated [50,51]. For the first time, we revealed enhanced LEPR levels in HBCs from PE vs. control placentas. This is in contrast to Li et al. [50], who reported no significant difference in LEPR levels in whole placental specimens obtained from PE vs. control placentas and LEPR immunoreactivity was detected in syncytiotrophoblasts, cytotrophoblasts, villous endothelial cells, Hofbauer cells, and endometrial decidual cells. This contradiction could be explained by primarily focusing on HBC-specific LEPR expressions between groups. Furthermore, we also showed significantly higher IL-6 immunoreactivity in HBCs of the PE vs. GA-matched control placentas. This finding complements our previous study that demonstrated enhanced IL-6 levels in both decidual cells and interstitial cytotrophoblasts at the maternal-fetal interface from PE placentas vs. the control [52]. Previous studies also demonstrated higher IL-6 immunoreactivity in syncytiotrophoblasts, endothelial cells, and macrophages of preeclamptic placenta vs. the control [53,54,55]. Overall, HSCORE results revealed the co-localization of IL-6 and LEPR in HBCs and their enhanced levels in PE placentas, suggesting a local pro-inflammatory action of leptin in HBCs. These findings also suggest the existence of a potential, previously undisclosed paracrine interaction in which trophoblasts generated leptin by binding to LEPR expressed in HBCs, which induces IL-6 expression, and then targeted neighboring trophoblasts to further promote inflammation in PE (Figure 6).

HBCs are fetal origin macrophages residing in the placental villi from as early as 4 weeks post-conception until birth [56,57]. HBCs typically display an anti-inflammatory M2 or M2-like phenotype [58]. Moreover, previous studies reported that HBCs display a switch from anti-inflammatory M2 to a proinflammatory M1 phenotype in patients with chorioamnionitis or villitis, or type 1 diabetes mellitus [59,60,61]. However, several reports indicate that despite expressing M2 markers, HBCs still express proinflammatory cytokines in response to bacterial or viral infection [42,62,63]. Similarly, Tang et al. reported a decreased expression of fetal macrophage (HBCs) markers such as CD163+, CD68+, and folate receptor β, and a reduced number of HBCs in placentas from pregnancies with severe PE [64]. Our group also confirmed the reduction of HBC numbers (Appendix A) in PE placentas [65]. These findings suggest that HBCs promote anti-inflammatory, pro-angiogenic responses in the normal placenta, while changes in the HBCs phenotype toward a proinflammatory phenotype and/or reduced numbers in PE placenta likely contribute to the dysregulation of normal placental development and function.

Our results support previous studies suggesting that leptin acts as an immune modulator, which regulates the production of several cytokines, both in vivo and in vitro, in the placenta as well as several other tissues [5,6,36,66,67]. Moreover, it has been reported that IL-6 induces leptin secretion in a concentration-dependent manner in first-trimester cytotrophoblast cultures [68]. Collectively, these results indicate that there is a bidirectional regulation of leptin and IL-6 in the placenta. We found that *IL-6* mRNA and protein levels in HBCs were increased by higher concentrations of leptin (100, 250, and 1000 ng/mL), but not by a lower leptin concentration (10 ng/mL). These in vitro results are consistent with in vivo results obtained from healthy pregnant women, in whom mean maternal serum leptin levels at term were reported to be 37.17 ± 28.1 ng/mL with the established reference range as 33.19–41.14 ng/mL [69,70]. However, in PE-complicated pregnancies, two-to-three-fold elevated leptin levels in maternal serum have been reported ranging between 80 and 90 ng/mL leptin concentrations in PE cases [23,24]. Two previous studies reported serum leptin levels ~230–260 ng/mL in PE patients [35,47]. Mise et al. also reported ~10-fold higher placental leptin mRNA levels in severe PE women compared to normal women [35]. Thus, leptin concentrations in the placenta have been noted to be significantly (more than 10-fold) higher than maternal and fetal serum concentrations, indicating a stronger local impact of leptin action in placental cells expressing LEPR [51,71,72,73,74,75]. Halleux et al. [75] showed that placental leptin secretion in vivo can reach levels greater than 1000 ng/mL. Thus, leptin concentrations used in this study represent both maternal serum and local placental leptin levels in inflammatory conditions.

In addition to evaluating the pro-inflammatory effects of leptin on HBCs, we also assessed the potential signaling pathways that regulate these effects. It is well established that LEPRs mediate leptin action mainly via the JAK/STAT pathway [3], as well as other common pathways such as NF-κB and ERK1/2 [5]. Our immunoblotting results showing a concentration-dependent significant increase in activation levels of STAT5, p65 NF-κB, and ERK1/2 after 24 h of leptin treatment suggest that the leptin-induced activation of these signaling cascades contributes to the induction of IL-6 expression in HBCs. This suggestion was further confirmed by treating HBCs with the specific inhibitors of each pathway that led to reduced *IL-6* mRNA levels. Contrary to common knowledge indicating that LEPRs act through JAK/STATs [3], Cauzac et al. [76] demonstrated that leptin does not activate either STAT3 or STAT1 signaling cascades, but activates the ERK1/2 cascade for the survival of BeWo cells, a placental choriocarcinoma cell line. Similarly, Perez-Perez et al. [20] using another trophoblastic cell line, JEG3 cells, reported that the anti-apoptotic effect of leptin is primarily regulated by the ERK1/2 pathway, but not the PI3K cascade, since this effect was completely reversed by blocking ERK1/2 activity with PD98059, whereas it was not affected by PI3K inhibition using wortmannin. In the current study, using similar strategies, we suggest that leptin-induced IL-6 regulation in HBCs is mediated primarily by the ERK1/2 pathway. According to our immunoblotting results, ERK1/2 phosphorylation displayed a different pattern than the other signaling pathways, since total ERK1/2 levels were also affected by leptin treatment, whereas the total STAT3, STAT5, and p65 NF-κB levels were unchanged. Furthermore, following treatment with the ERK1/2 inhibitor PD98059, *IL-6* mRNA levels were subsequently decreased by ~70% at 250 ng/mL and 1000 ng/mL leptin concentrations, whereas incubation with other inhibitors was able to reduce *IL-6* mRNA levels only by ~30% (STAT inhibitors) and ~50% (NF-κB inhibitor). Thus, these findings revealed that the ERK1/2 pathway represents the primary signaling cascade regulating leptin action in HBCs.

Taken together, our findings suggest that HBCs exert pro-inflammatory effects at least in part through excess leptin-LEPR signaling to serve as one of the primary sources of IL-6 production in PE women. This augmented expression of IL-6 by fetal macrophages may play a crucial role in the pathogenesis of PE by promoting endothelial cell dysfunction in view of their proximity to vascular structures in placental villi; and indirectly by impeding trophoblast invasion by contributing to the excess pro-inflammatory environment of PE.

## 4. Materials and Methods

### 4.1. Placental Tissue Collection

Collection of placental specimens was approved by the University of South Florida Institutional Review Board (#Pro00019472). After receiving written informed consent, placental specimens were obtained from deidentified patients with GA-matched pregnancies obtained from control (n = 7) or with PE (n = 8). Criteria for PE diagnosis included a blood pressure > 140 mmHg systolic or >90 mmHg diastolic, accompanied by proteinuria after 20 weeks of gestation, as we described [77,78] using the ACOG definition of PE [79,80]. Moreover, clinical information is provided in Table 1.

### 4.2. Immunohistochemistry

Paraffin sections from placental specimens obtained from GA-matched control and PE pregnancies were deparaffinized and rehydrated. Heat-induced antigen retrieval using 10 mM citrate buffer (pH 6.0) was followed by endogenous peroxidase quenching with 3% hydrogen peroxide for 15 min and blocking with 5% normal horse serum (Vector Labs, Newark, CA, USA) for 30 min at room temperature. Sections were then incubated overnight at 4 °C with either a goat IL-6 (R&D Systems, Minneapolis, MN, USA) or mouse LEPR antibody (Santa Cruz Biotechnology Inc., Santa Cruz, CA, USA) that recognizes all short and long isoforms of LEPR. After several washes with Tris-buffered saline (TBS, pH 7.2), the slides were incubated with corresponding biotinylated secondary antibodies (Vector Labs) for 30 min at room temperature, then with streptavidin-biotin complex kit (ABC Kit, Vector Labs) for 30 min. Immunoreactivity was developed using chromogen diaminobenzidine (DAB; 3,3′-diaminobenzidine tetrahydrochloride dihydrate; Vector Labs) and the sections were lightly counterstained using Mayer’s hematoxylin (Vector Labs). To identify HBCs, a second set of serial sections of placental specimens was also immunostained with a mouse CD68 antibody (a macrophage marker, Santa Cruz). For negative controls, appropriate normal IgG isotypes were used at the same concentration of each corresponding primary antibody. Immunoreactivity for IL-6 and LEPR was assessed by histological scoring (HSCORE), a semi-quantitative method that evaluates the intensity, and the number of immunostained cells was performed by two blinded investigators, as described [81] using the Axio Imager-A2 light microscope (Zeiss; Oberkochen, Germany) and the ZEN imaging system (Zeiss).

### 4.3. Isolation and Culture of Hofbauer Cells

HBCs (n = 3) were isolated from uncomplicated term placental specimens, as described previously [82]. Briefly, villous tissues were sequentially exposed to a trypsin/DNase I, then a collagenase A/DNase I enzyme mixture. Resuspended cell mixtures were loaded onto a discontinuous Percoll gradient (35/30/25/20%). Cell suspension from 20–25% to 30–35% interfaces were combined and immunopurified by negative selection using sequential treatment with the DynaBeads conjugated anti-epidermal growth factor receptor and then anti-CD10 antibodies (Santa Cruz and Biolegend, San Diego, CA, USA, respectively). After magnetic separation, purified cells were plated in 6-well culture plates. Floating and weakly adherent cells were washed and discarded after 1 h incubation. The remaining adherent cells were cultured in DMEM/F12 medium (Life Technologies, Grand Island, NY, USA) containing 10% FBS and 1% antibiotic-antimycotic (Life Technologies).

#### 4.3.1. Experimental Setup

HBCs seeded into 6-well culture dishes were incubated with either vehicle control or recombinant human leptin (r-leptin, #300-27-IMG, PeproTech Inc., East Windsor, NJ, USA) at 10, 100, or 1000 ng/mL concentrations in serum-free DMEM/F12 medium supplemented with 1% ITS+ premix (Corning Inc., Corning, NY, USA) for 6 h for mRNA or 24 h for protein analysis. To rule out the potential endotoxin contamination of r-leptin on IL-6 levels, cells were pre-incubated with 1000 ng/mL lipopolysaccharides (LPS) antagonist (LPS-RS; Invivogen, San Diego, CA, USA) for 1 h, followed by 1 ng/mL LPS (L2880, Sigma-Aldrich, Burlington, MA, USA) or 1000 ng/mL r-leptin incubations for 6 h.

To identify relevant signaling pathways involved in leptin-mediated IL-6 expression, HBCs were pre-treated with either control or 1 µM STAT3 inhibitor III, or 50 µM STAT5 inhibitor, 10 µM NF-κB activation inhibitor III, or 20 µM PD98059 as an ERK1/2 inhibitor for 1 h, then treated with either control or 100, or 250, or 1000 ng/mL r-leptin for 6 h for qPCR analysis. All inhibitors were purchased from Millipore (San Diego, CA, USA).

#### 4.3.2. Western Blot Analysis

To determine leptin-induced IL-6 levels as well as leptin-mediated signaling pathway(s) of IL-6 expression, HBC lysates (n = 3) were subjected to SDS-PAGE with Tris-HCl gels (Bio-Rad, Hercules, CA, USA), then transferred on to a nitrocellulose membrane. Following blocking with 10% non-fat dried milk, the membrane was incubated overnight with IL-6 antibody or phosphorylated (ph-) or total (t-) antibodies against STAT5, STAT3, p65 NF-κB, and ERK1/2 MAPK (Table 2). After washing with TBS-T (TBS with 0.1% Tween 20; pH:7.2), the membrane was incubated with horseradish peroxidase (HRP)-conjugated secondary antibodies (Vector Labs) for 30 min at room temperature followed by development of immunoblotting signals using a chemiluminescence kit (GE Healthcare, Piscataway, NJ, USA). All membranes were stripped and re-probed with HRP-labelled rabbit anti-β-actin antibody (Cell Signaling, Danvers, MA, USA).

#### 4.3.3. Enzyme-Linked Immunosorbent Assay (ELISA)

Secreted IL-6 levels in the conditioned media obtained from 24 h leptin-treated HBC cultures (n = 3) were measured by an ELISA kit (#DY206, R&D Systems) according to the manufacturer’s instructions.

#### 4.3.4. Quantitative Real-Time PCR

Total RNA extraction was performed using the miRNeasy kit (Qiagen, Valencia, CA, USA) according to the manufacturer’s instructions. Then, 500 ng total RNA was reverse transcribed using the Retroscript kit (Invitrogen, Carlsbad, CA, USA) with random decamer primers at 42 °C for 1 h. Expression of *IL-6* mRNA levels was analyzed by using Taqman Gene Expression assay (Applied Biosystems, Foster City, CA, USA) and normalized to β–actin mRNA levels. All samples (n = 3) were run in duplicates and calculated using the 2^−ΔΔCt^ formula.

### 4.4. Statistical Analysis

For two-group comparisons, *t*-test (for parametric data) or Mann–Whitney Rank Sum test (for non-parametric data) were performed. For multiple comparisons, Kruskal–Wallis One Way Analysis of Variance on Ranks was performed, followed by Student–Newman–Keuls multiple comparison method (for non-parametric data). Statistical significance was defined as *p*  <  0.05. Statistical analysis was performed using Sigmaplot version 11.0 (Systat Software, Inc, San Jose, CA, USA).

## Figures and Tables

**Figure 1 ijms-25-00135-f001:**
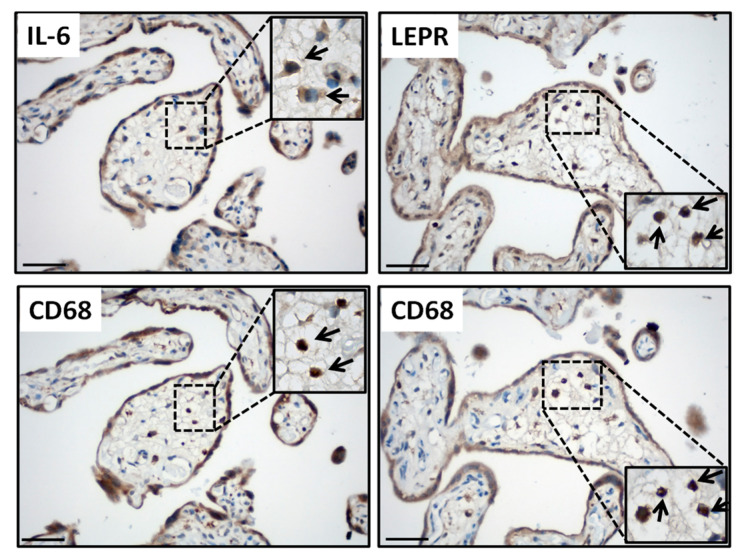
Hofbauer cells display strong IL-6 and leptin receptor expression in placenta. Representative IL-6 and leptin receptor (LEPR) immunoreactivity (**upper panels**) in Hofbauer cells that are immunostained with CD68 (macrophage marker) in serial sections of control placentas (**lower panels**). Arrows indicate IL-6 or LEPR-expressing CD68-positive Hofbauer cells. Scale bar = 30 µm. Insets represent the higher magnification of relevant areas.

**Figure 2 ijms-25-00135-f002:**
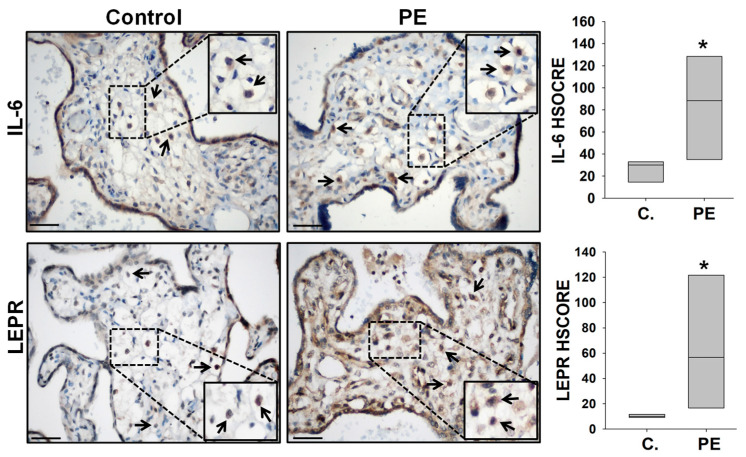
IL-6 and LEPR immunostaining in control and preeclamptic placentas. Representative images of IL-6 and LEPR immunostaining in preeclamptic (PE) and gestational aged-matched control placentas. Hofbauer cells-specific HSCORE analysis demonstrated significantly higher IL-6 (n = 5/group) and LEPR expression in PE placentas (n = 8) vs. gestational aged-matched controls (n = 7) (83.0 ± 20.9 vs. 25.0 ± 4.5 and 63.5 ± 18.6 vs. 10.9 ± 1.0, respectively). Arrows indicate IL-6 and LEPR-expressing Hofbauer cells. Bars represent median values, * *p* < 0.05 vs. control (C) specimens analyzed by Mann–Whitney Rank Sum test. Scale bar = 30 µm.

**Figure 3 ijms-25-00135-f003:**
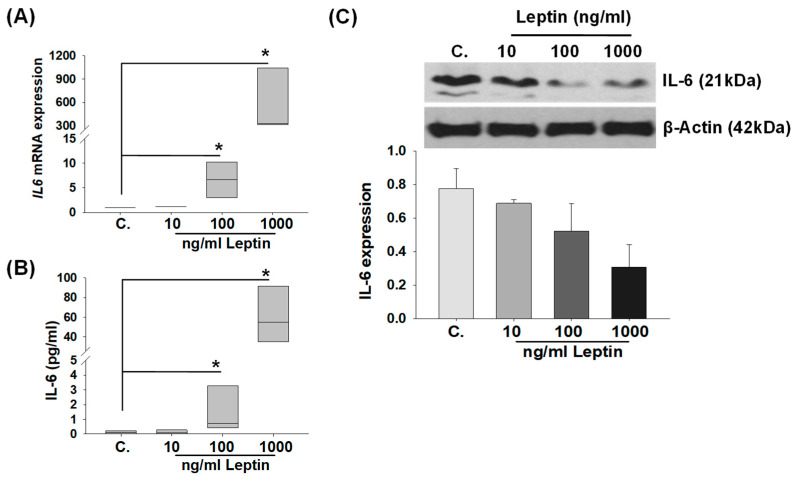
Leptin-induced IL-6 levels in primary cultured Hofbauer cells. (**A**) *IL-6* mRNA levels in Hofbauer cells after 6 h r-leptin treatment (10, 100, and 1000 ng/mL) by qPCR. Data represent fold change as median values; * *p* < 0.05 vs. control (C) vehicle-treated group (n = 3) analyzed by Kruskal–Wallis One Way ANOVA on Ranks followed by Student–Newman–Keuls multiple comparison. (**B**) Leptin-induced secreted IL-6 levels (n = 4 from 3 different patients) in conditioned media supernatant obtained from Hofbauer cell culture after 24 h r-leptin treatment (10, 100, and 1000 ng/mL) by ELISA. Bars represent median values; * *p* < 0.05 vs. control (C) vehicle-treated group analyzed by Mann–Whitney Rank Sum test. (**C**) Immunoblot analysis of Hofbauer cell lysates after 24 h displaying reduced IL-6 expression in 100 and 1000 ng/mL leptin-treated cells. Bars represent mean ± SEM values from IL-6 protein levels after being normalized to β-actin, analyzed by Kruskal–Wallis One Way Analysis of Variance on Ranks, followed by the Student–Newman–Keuls multiple comparison method.

**Figure 4 ijms-25-00135-f004:**
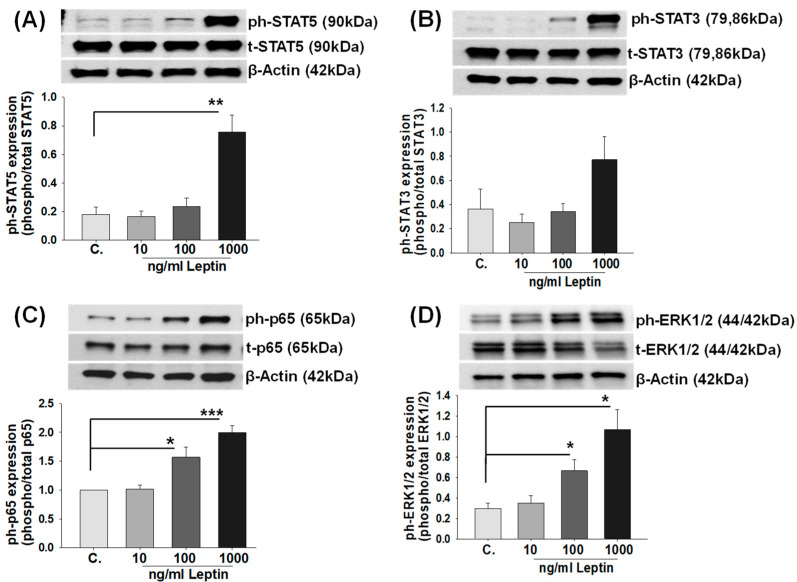
Leptin-mediated activation of STAT3, STAT5, NF-κB p65, and ERK1/2 signaling pathways in Hofbauer cells. Representative immunoblotting of Hofbauer cells treated with either control (C) or r-leptin (10, 100, 100 ng/mL) displays phosphorylated (ph) and total (t) levels of (**A**) STAT5; (**B**) STAT3; (**C**) p65 NF-κB; and (**D**) ERK1/2 MAPK, as well as β-actin. Graphs display immunoblot densitometry readings obtained from experimental incubations of cells for phosphorylated/total forms of each protein after being normalized to β-actin. Bars represent mean ± SEM, n = 3, * *p* < 0.05, ** *p* = 0.01 or *** *p* = 0.001 vs. vehicle-treated control (C) analyzed by *t*-test.

**Figure 5 ijms-25-00135-f005:**
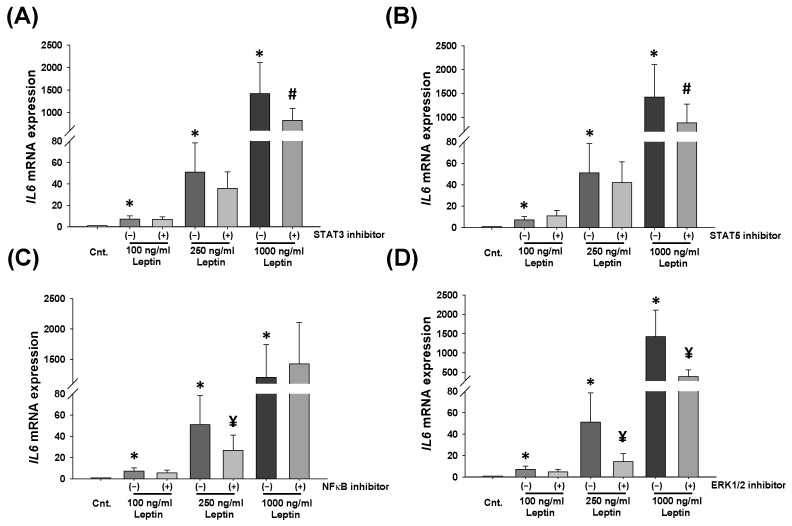
Leptin-induced *IL-6* mRNA expression in Hofbauer cells is primarily mediated by ERK1/2 signaling pathway. *IL-6* mRNA levels in Hofbauer cells treated with 100, 250, and 1000 ng/mL r-leptin ± specific inhibitors for STAT3 (**A**), STAT5 (**B**), p65 NF-κB (**C**), and ERK1/2 (**D**) signaling pathways. Bars represent mean ± SEM, n = 3, * *p* < 0.05 vs. control (Cnt), analyzed by Kruskal–Wallis One Way Analysis of Variance on Ranks followed by Student–Newman–Keuls multiple comparison method, # *p* < 0.05 leptin with inhibitor vs. without inhibitor (**A**,**B**), ¥ *p* ≤ 0.001 leptin with inhibitor vs. without inhibitor (**C**,**D**) analyzed by *t*-test.

**Figure 6 ijms-25-00135-f006:**
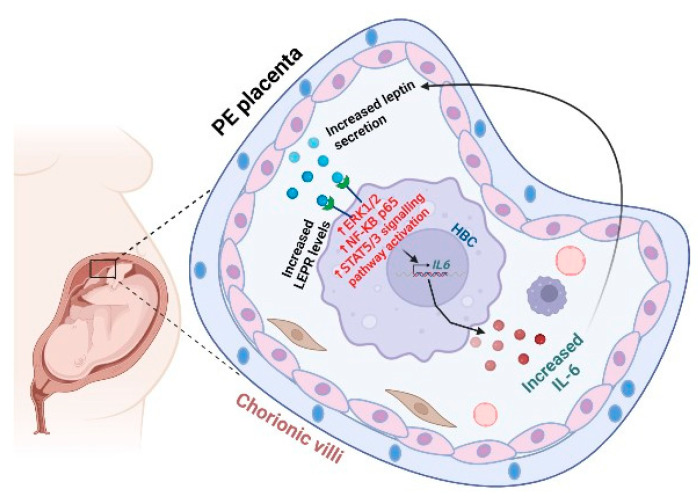
Bidirectional crosstalk between Hofbauer cells and leptin-expressing trophoblasts induces IL-6 levels in HBCs in preeclamptic placenta. Preeclamptic (PE) placentas display increased leptin expression generated primarily from syncytiotrophoblasts, extravillous trophoblasts, as well as decidual stromal cells, and leptin receptor (LEPR) levels in Hofbauer cells (HBCs). Secreted leptin (blue circle) binding to LEPR activates ERK1/2 MAPK, p65 NF-κB, and STAT 3/5 signaling pathways to induce IL-6 (brown circle) expression in HBCs, which then may further promote leptin secretion in PE placentas, which causes a vicious cycle of leptin-triggered inflammation. Upon induction by excess leptin, HBCs may serve as one of the primary sources of IL-6 production in PE women.

**Table 1 ijms-25-00135-t001:** Patients’ clinical data that are used in this study.

	Control (Mean ± SD; n = 7)	Preeclampsia (Mean ± SD; n = 8)	*p* Value
**Gestational Age**	38.15 ± 1.15	36.73 ± 1.72	*p* = 0.089
**BMI**	34.57 ± 13.47	34.12 ± 1.1	*p* = 0.232
**Maternal Age**	32.714 ± 5.99	28.62 ± 4.59	*p* = 0.159
**Birth Weight**	3415.00 ± 507.08	2670.62 ± 621.43	*p* = 0.026

**Table 2 ijms-25-00135-t002:** Antibodies used in this study. WB: Western blotting, IHC: Immunohistochemistry.

Antibody	Company and Catalog Number	Application Dilution
Phospho-ERK1/2	Cell Signaling #4370.	WB 1:1000
Total-ERK1/2	Cell Signaling #9102	WB 1:1000
Phospho-STAT3	Cell Signaling #9145	WB 1:1000
Total-STAT3	Cell Signaling #9139	WB 1:1000
Phospho-STAT5	Cell Signaling #9359	WB 1:1000
Total-STAT5	Cell Signaling #94205	WB 1:1000
Phospho-p65	Cell Signaling #3033	WB 1:1000
Total-p65	Cell Signaling #8242	WB 1:1000
IL-6	Santa Cruz #sc-28343	WB 1:1000
Β-Actin	Cell Signaling #5125	WB 1:1000
IL-6	R&D #AF-206-NA	IHC: 10 μg/mL
LepR (ObR)	Santa Cruz #sc-8391(B-3)	IHC 1:100
CD68	Santa Cruz #sc-20060 (KP1)	IHC 1:1000

## Data Availability

Data supporting results of the current study are available from the corresponding author upon request.

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
