# Peer review of "Leptin-Mediated Induction of IL-6 Expression in Hofbauer Cells Contributes to Preeclampsia Pathogenesis"

_ijms, 2023, doi:10.3390/ijms25010135_

Round 1
Reviewer 1 Report
Comments and Suggestions for Authors
Why was Leptin elevated in PE placenta? What were the initiating factors?
Title 2.1 is missing.
Discussion: “IL-6 mRNA and protein levels in HBCs were increased by higher concentrations of leptin.” Why were the trends in Figure 3B and Figure 3C different?
Whether IL-6 produced by other cell types also has an effect on PE?
Author Response
Dear Dr. Raylene Zou;
Thank you and the reviewers for the prompt and comprehensive review of our manuscript (ijms#2699847) entitled “Leptin Mediated Induction of IL-6 Expression in Hofbauer Cells Contributes to Preeclampsia Pathogenesis” We appreciate the reviewers’ and editor’s comments and respond to them point by point as follows:
Responses to Comments by Reviewer 1:
Q1. Why was Leptin elevated in PE placenta? What were the initiating factors?
Response 1: Several studies reported that increased hypoxia, oxidative stress, and pro-inflammatory cytokines contribute to enhanced levels of leptin in preeclamptic placenta [1-4]. As suggested by the reviewer, this statement is included in the Discussion of the revised manuscript (Lines 224-226).
Q2. Title 2.1 is missing.
Response 2: The title 2.1 ‘Elevated IL-6 and LEPR Immunoreactivity in Hofbauer Cells of PE Placentas’ is now added to the results section (Line 93).
Q3: Discussion: “IL-6 mRNA and protein levels in HBCs were increased by higher concentrations of leptin.” Why were the trends in Figure 3B and Figure 3C different?
Response 3: Our findings revealed that treatment of HBCs with 100 and 1000 ng/ml Leptin increases IL-6 mRNA expression (Fig. 3A) and secretion (Fig. 3B) while decreasing intracellular protein levels of IL-6. This suggests that lower intracellular IL-6 protein levels are due to mobilization of IL-6 for secretion from HBCs after leptin exposure. This statement was already mentioned in the manuscript (Lines 150-151).
Q4: Whether IL-6 produced by other cell types also has an effect on PE?
Response 4: We previously reported that IL-6 is predominantly expressed by decidual stromal cells and interstitial trophoblasts at the maternal-fetal interface and these cells exhibit significantly higher levels of IL-6 in preeclamptic placentas compared to gestational-age matched control. These findings and related citation are presented in this manuscript (Lines 240-242). Additionally, “previous studies also demonstrated higher IL-6 immunoreactivity in syncytiotrophoblasts, endothelial cells and macrophages of preeclamptic placenta vs. control [5-7]” These studies are now included in the revised manuscript (Lines 242-244).
Reviewer 2 Report
Comments and Suggestions for Authors
Go through the text carefully and correct the mistakes!
e.g. 2. Results row is missing; Figure 3A IL6 corrected: IL-6 The same error in the text.
1. I recommend testing the polarization of HBCs samples with M1 and M2 markers (CD80 and CD206).
2. It would have been useful to investigate how changing the polarization (treatment with LPS+ IFNgamma (M1 polarization) and IL-4 (M2 polarization) ) modifies the results? Unfortunately, this can no longer be done.
3. Were the GA control pregnant women healthy? Based on the data, it is likely that a premature birth may have occurred.
4. Clinical data are missing (mother's age, BMI, systolic and diastolic blood pressure, proteinuria, odema, laboratory inflammatory markers, birth weight (intrauterine growth retardation)). These data need to be replaced! If you make a table, include the week of the term! Were preeclamptic cases classified as severe or mild?
5. HBC primary cell culture was prepared from which samples of severe PE patients?
6. SEM cannot be used for small sample numbers (n=3), calculate SD!
Author Response
Dear Dr. Raylene Zou;
Thank you and the reviewers for the prompt and comprehensive review of our manuscript (ijms#2699847) entitled “Leptin Mediated Induction of IL-6 Expression in Hofbauer Cells Contributes to Preeclampsia Pathogenesis” We appreciate the reviewers’ and editor’s comments and respond to them point by point as follows:
Responses to Comments by Reviewer 2:
Q1: Go through the text carefully and correct the mistakes! e.g.2. Results row is missing; Figure 3A IL6 corrected: IL-6 The same error in the text.
Response 1: Thank you for the Reviewer’s comments. We carefully revised and corrected typing errors as suggested. These include: 1) The title of “Results 2.1” is now included as “Elevated IL-6 and LEPR Immunoreactivity in Hofbauer Cells of PE Placentas” (Line 93); and 2) IL6 is replaced with IL-6 throughout manuscript and in Figure 3A.
Q2: I recommend testing the polarization of HBCs samples with M1 and M2 markers (CD80 and CD206).
Response 2: In the current study, M2 to M1 polarization of HBCs is not tested since others have previously studied this phenomenon and their findings were discussed in the original manuscript (see references 42, 58-64) (Lines 261-269). Collectively, the literature suggests that HBCs do not fully polarize toward either a distinct M1 or M2 phenotype, and display characteristics of both phenotypes in PE.
Q3: It would have been useful to investigate how changing the polarization (treatment with LPS+ IFNgamma (M1 polarization) and IL-4 (M2 polarization) modifies the results? Unfortunately, this can no longer be done.
Response 3: Schliefsteiner et al. (Ref #63,) isolated HBCs from healthy term placentas, treated them with LPS and INF-γ or IL-4 and IL-13 to induce the M1 and M2 phenotype, respectively. Specific cell polarization markers and cytokines, associated with respective phenotypes, were investigated by flow cytometry and ELISA. Results showed that proinflammatory stimuli reduced M2 markers but did not induce M1 markers. TNF-α release was increased, but at the same time TGF-β and IL-10 release was also upregulated, resembling in part the M2b sub-phenotype. Anti-inflammatory stimuli with IL-4 and IL-13 displayed no effect on HBC polarization, indicating that HBCs maintain their M2 phenotype in vitro under inflammatory stimuli.
Q4: Were the GA control pregnant women healthy? Based on the data, it is likely that a premature birth may have occurred.
Response 4: Gestational age matched control placentas were collected from uncomplicated healthy patients. This clinical data is now given in Table 1 in the revised manuscript (Lines 338-346).
Q5: Clinical data are missing (mother's age, BMI, systolic and diastolic blood pressure, proteinuria, odema, laboratory inflammatory markers, birth weight (intrauterine growth retardation). These data need to be replaced! If you make a table, include the week of the term! Were preeclamptic cases classified as severe or mild?
Response 5: Criteria for PE diagnosis included a blood pressure >140 mmHg systolic or >90 mmHg diastolic accompanied by proteinuria after 20 weeks of gestation as we described previously [8,9] (Lines 333-335). We did not classify PE cases as severe or mild. As suggested by Reviewer, Table 1 (Lines 338-346) now provides clinical data including gestational age, maternal age, BMI and birth weight in the revised manuscript.
Q6: HBC primary cell culture was prepared from which samples of severe PE patients?
Response 6: HBCs were isolated from uncomplicated term placental specimens as mentioned in Material and Methods section (Lines 371-372).
Q7: SEM cannot be used for small sample numbers (n=3), calculate SD!
Response 7: We used Sigmaplot Version 11.0, Systat Software, Inc, for our statistical analysis and calculated both SEM and SD when the samples were normally distributed with an equal variance. The SD is an indication of the dispersion of the data, whereas the SE is an indication of the dispersion (or precision) or an estimate (like the sample mean, for instance). One can easily calculate SD values using data given in the manuscript. Thus, we would prefer to keep our graphs and data with SEM values consistent with our 30-year publications record.
References:
- Laivuori, H.; Gallaher, M. J.; Collura, L.; Crombleholme, W. R.; Markovic, N.; Rajakumar, A.; Hubel, C. A.; Roberts, J. M.; Powers, R. W., Relationships between maternal plasma leptin, placental leptin mRNA and protein in normal pregnancy, pre-eclampsia and intrauterine growth restriction without pre-eclampsia. Molecular human reproduction 2006, 12, (9), 551-6.
- Mise, H.; Sagawa, N.; Matsumoto, T.; Yura, S.; Nanno, H.; Itoh, H.; Mori, T.; Masuzaki, H.; Hosoda, K.; Ogawa, Y.; Nakao, K., Augmented placental production of leptin in preeclampsia: possible involvement of placental hypoxia. The Journal of clinical endocrinology and metabolism 1998, 83, (9), 3225-9.
- Redman, C. W.; Sargent, I. L., Placental stress and pre-eclampsia: a revised view. Placenta 2009, 30 Suppl A, S38-42.
- Sitras, V.; Paulssen, R. H.; Gronaas, H.; Leirvik, J.; Hanssen, T. A.; Vartun, A.; Acharya, G., Differential placental gene expression in severe preeclampsia. Placenta 2009, 30, (5), 424-33.
- Aggarwal, R.; Jain, A. K.; Mittal, P.; Kohli, M.; Jawanjal, P.; Rath, G., Association of pro- and anti-inflammatory cytokines in preeclampsia. J Clin Lab Anal 2019, 33, (4), e22834.
- Gray, G.; Scroggins, D. G.; Wilson, K. T.; Scroggins, S. M., Cellular Immunotherapy in Mice Prevents Maternal Hypertension and Restores Anti-Inflammatory Cytokine Balance in Maternal and Fetal Tissues. Int J Mol Sci 2023, 24, (17).
- Ma, Y.; Ye, Y.; Zhang, J.; Ruan, C. C.; Gao, P. J., Immune imbalance is associated with the development of preeclampsia. Medicine (Baltimore) 2019, 98, (14), e15080.
- Canfield, J.; Arlier, S.; Mong, E. F.; Lockhart, J.; VanWye, J.; Guzeloglu-Kayisli, O.; Schatz, F.; Magness, R. R.; Lockwood, C. J.; Tsibris, J. C. M.; Kayisli, U. A.; Totary-Jain, H., Decreased LIN28B in preeclampsia impairs human trophoblast differentiation and migration. FASEB J 2019, 33, (2), 2759-2769.
- Ozmen, A.; Guzeloglu-Kayisli, O.; Tabak, S.; Guo, X.; Semerci, N.; Nwabuobi, C.; Larsen, K.; Wells, A.; Uyar, A.; Arlier, S.; Wickramage, I.; Alhasan, H.; Totary-Jain, H.; Schatz, F.; Odibo, A. O.; Lockwood, C. J.; Kayisli, U. A., Preeclampsia is Associated With Reduced ISG15 Levels Impairing Extravillous Trophoblast Invasion. Front Cell Dev Biol 2022, 10, 898088.
Round 2
Reviewer 1 Report
Comments and Suggestions for Authors
None